# Expression of MALAT1 Promotes Trastuzumab Resistance in HER2 Overexpressing Breast Cancers

**DOI:** 10.3390/cancers12071918

**Published:** 2020-07-16

**Authors:** Yanyuan Wu, Marianna Sarkissyan, Ochanya Ogah, Juri Kim, Jaydutt V. Vadgama

**Affiliations:** 1Division of Cancer Research and Training, Department of Medicine, Charles R. Drew University of Medicine and Science, 1731 East 120th Street, Los Angeles, CA 90059, USA; mariansark@gmail.com (M.S.); ochanya.ogah@gmail.com (O.O.); jurik@uci.edu (J.K.); 2Jonsson Comprehensive Cancer Center, David Geffen School of Medicine, University of California at Los Angeles, Los Angeles, CA 90095, USA

**Keywords:** MALAT1, FOXO1, breast cancer, trastuzumab resistance

## Abstract

Background: Metastasis-associated lung adenocarcinoma transcript 1 (MALAT1) is associated with cancer progression. Our study examined the role of MALAT1 in breast cancer and the mechanisms involved in the regulation of MALAT1. Methods: In vitro cell and in vivo animal models were used to examine the role of MALAT1 in breast cancer. The interaction of FOXO1 (Forkhead Box O1) at the promoter region of MALAT1 was investigated by chromatin immunoprecipitation (ChIP) assay. Results: The data shows an elevated expression of MALAT1 in breast cancer tissues and cells compared to non-cancer tissues and cells. The highest level of MALAT1 was observed in metastatic triple-negative breast cancer and trastuzumab-resistant HER2 (human epidermal growth factor receptor 2) overexpressing (HER2+) cells. Knockdown of MALAT1 in trastuzumab-resistant HER2+ cells reversed epithelial to mesenchymal transition-like phenotype and cell invasiveness. It improved the sensitivity of the cell’s response to trastuzumab. Furthermore, activation of Akt by phosphorylation was associated with the upregulation of MALAT1. The transcription factor FOXO1 regulates the expression of MALAT1 via the PI3/Akt pathway. Conclusions: We show that MALAT1 contributes to HER2+ cell resistance to trastuzumab. Targeting the PI3/Akt pathway and stabilizing FOXO1 translocation could inhibit the upregulation of MALAT1.

## 1. Introduction

Therapeutic failure and distant metastasis have been significant challenges in the treatment of breast cancer as well as the leading cause of mortality in breast cancer patients. Compared to all different types of breast cancers, human epidermal growth factor receptor 2 (HER2), overexpressing (HER2+) and triple-negative breast cancers (TNBC, estrogen/progesterone receptors negative (ER/PR-) and HER2-) are more likely to develop the metastatic disease due to their aggressive tumor characteristics [1,2,3,4]. Treatment with anti-HER2 antibody (Herceptin^®^, also known as trastuzumab) has made a significant difference in the overall survival rates. However, de novo and acquired resistance to trastuzumab is a considerable challenge for 52% of HER2+ breast cancer patients receiving trastuzumab treatment [5,6]. TNBC type tumors are often aggressive and have a poorer prognosis due to limited treatment options [3,4]. Hence we must improve our understanding of the disease and develop new biomarkers and strategies for cancer treatment.

Previous studies on mechanisms of tumor metastases have focused on protein-coding genes. Still, recently it has been recognized that tumor metastases also involve non-translated genes [7]. Protein-coding genes account only for 2% of the human genome. Non-coding RNAs (ncRNAs) can be categorized into two major classes based on their size: small ncRNAs with <200 nt in length, such as micro RNAs (miRNAs) and long ncRNAs (lncRNAs) with >200 nt in length [8]. Emerging evidence has shown that lncRNAs can play a critical role in tumorigenesis or cancer progression by regulating gene expression through various mechanisms such as regulation of transcription, translation, protein modification, and formation of RNA-protein complexes [7,9]. LncRNAs could also interact with cell signaling pathways in cancer cells and mediate tumor progression [10].

Among those lncRNAs, metastasis-associated lung adenocarcinoma transcript 1 (MALAT1), has been implicated in cancer signaling pathways. Also known as non-coding nuclear-enriched abundant transcript 2 (NEAT2), MALAT1 was initially identified as a prognostic marker for metastasis and poor patient survival in non-small cell lung carcinoma (NSCLC) [11]. Recent studies showed that MALAT1 is upregulated in several solid tumors, including lung, prostate, colon, pancreas, cervical, and liver cancers. MALAT1 is an adverse prognostic factor [12,13,14,15,16,17,18]. High expression of MALAT1 in breast cancer cases and may be associated with triple-negative breast cancer (TNBC) [19,20]. The upregulation of MALAT1 may play an essential role in breast cancer development and associated with lymph node metastasis [21]. Using the combined data from eight Gene Expression Omnibus (GEO) datasets, plus the Cancer Genome Atlas (TCGA) breast cancer provisional data plus in their study, Wang et al., found that high expression of MALAT1 in breast cancer is associated with reduced relapse-free survival [22]. MALAT1 could serve as a predicting and prognosis marker for breast cancer progression [23,24]. Knockdown of MALAT1 using antisense oligonucleotides (ASOs) in the mouse mammary tumor virus (MMTV)-PyMT mouse mammary carcinoma model results in slower tumor growth accompanied by significant differentiation into cystic tumors and a reduction in metastasis [25]. However, Kim et al., reported recently that knockdown of MALAT1 in MMTV-PyMT mouse increased mammary tumor metastasis [26]. Kim’s study also reported their data in TCGA, showing that the MALAT1 level was downregulated in human breast tumors compared to normal tissues. They also showed that MALAT1 expression decreased in metastatic tumors compared to primary tumors [26]. These discrepant findings encourage more studies to understand better the role of MALAT1 in breast cancer and the underlying molecular mechanisms associated with metastases.

Signaling pathways associated with MALAT1-induced cell proliferation and metastasis include the phosphatidylinositol-3-kinase/serine/threonine kinase (PI3K) and protein kinase B (Akt) pathway [27,28,29]. MALAT1 has been shown to regulate cell proliferation and cisplatin resistance via the PI3K/Akt pathway in cervical and gastric cancer [27,28,29]. It has also been reported to promote cell proliferation and epithelial-to-mesenchymal transition (EMT) via PI3K/AKT pathway in epithelial ovarian cancer [29]. In osteosarcoma, MALAT1 can promote cancer metastasis, mediated by activation of the PI3K/Akt signaling pathway [30,31], and the FOXO1-MALAT1-miR-26a-5p feedback loop [30,31].

The PI3K/Akt pathway is frequently altered in cancers and associated with treatment resistance in various tumor types [32,33,34]. Phosphorylation of Akt leads to dysregulation of its signaling pathway, subsequently inhibiting cell apoptosis and promoting cell survival [32,34]. Akt can be activated by phosphorylation at the plasma membrane, then translocated to the nucleus, or directly activated in the nucleus by nuclear pools of PI3K and phosphorylation by PDK1 [35,36,37]. Forkhead Box O1 (FOXO1), a member of the Forkhead transcription factor family, is one of the critical downstream mediators of the PI3K/AKT pathway [35,38,39]. Nuclear retention of FOXO1 protein plays a tumor-suppressive role in the prostate, breast, and soft tissue sarcoma [40,41,42]. Activation of the PI3K/Akt pathway phosphorylates FOXO1 and promotes the nuclear exclusion of FOXO1 [36,37]. We have shown previously that in HER2+ breast cancer cells, loss of nuclear expression FOXO1 by constitutive activation of Akt at Ser473 contributes to trastuzumab resistance [43]. Inhibition of pAkt restores the nuclear accumulation of FOXO1 and improves the response of HER2+ cells to trastuzumab treatment [43].

We designed this study to examine the role of MALAT1 in breast cancer and its association with HER2+ breast cancer cells resistant to trastuzumab. From ongoing studies, we identified MALAT1 in our gene array analysis. In breast cancer cells that were resistant to trastuzumab and/or tamoxifen, MALAT1 was one of the most significantly upregulated molecules, compared to cancer cells that were sensitive to treatment. Our results from this study show that MALAT1 induces EMT and cell invasion, and promotes HER2+ cells to become resistant to trastuzumab treatment. The knockdown of MALAT1 improves the sensitivity of HER2+ cells to trastuzumab and inhibits cell viability. FOXO1 has been reported as a transcriptional factor of MALAT1 that negatively regulates MALAT1 in osteosarcoma cells [30]. We found in this study that FOXO1 plays a regulatory role in mediating MALAT1 expression in breast cancer cells via the PI3/Akt pathway.

## 2. Results

### 2.1. MALAT1 Expression in Breast Cancer Tissue and Cell Lines

MALAT1 expression was examined in 13 breast cancer tissues and seven non-cancer tissues through mRNA-seq. The non-cancer tissue was the normal tissue adjacent to breast carcinoma from the same patient.

As shown in Figure 1A, the expression of MALAT1 was significantly higher in breast cancer tissues than that in non-cancer tissues. Furthermore, the MALAT1 expression was higher in lymph node positive breast tumors compared to node-negative tumors (Figure 1B). Similar results were observed by comparing breast cancer cell lines with non-cancer cell lines. Compared to non-breast cancer cells MCF12A, expression of MALAT1 was significantly upregulated in all subtypes of breast cancer cells (Figure 1C). Within breast cancer cell lines, the highest expression of MALAT1 was seen in metastatic TNBC cells, MB231 and trastuzumab-resistant HER2+ cells, JIMT1. The trastuzumab-resistant HER2+ cells SKBR3/100-8 and BT474/100-2 were generated from parental SKBR3 and BT474 through colony selection and confirmed resistance to trastuzumab [44]. Compared to their parental cell lines SKBR3 and BT474, the expression of MALAT1 was increased by 2.5-fold in SKBR3/100-8 and 2-fold in BT474/100-2 cells (Figure 1D). The data in Figure 1 suggests that MALAT1 is upregulated in all subtypes of breast cancer. However, the upregulation of MALAT1 is more significantly associated with HER2+ cells that are resistant to trastuzumab, and with metastatic TNBC in breast cancer cells.

### 2.2. Knockdown MALAT1 Reverses Trastuzumab Resistance in HER2+ Breast Cancer Cells

Next, we tested if the downregulation of MALAT1 could reverse the resistance to trastuzumab in HER2+ cells. SKBR3/100-8 and BT474/100-2 treated with siRNA MALAT1 showed significant downregulation of MALAT1 in SKBR3/100-8 and BT474/100-2 cells (Figure 2A). The SKBR3/100-8 and BT474/100-2 knockdown of MALAT1 showed a significant improvement to trastuzumab treatment. As shown in Figure 2B, trastuzumab treatment increased the inhibition of cell viability by almost 20% in siRNA-treated SKBR3/100-8 and BT474/100-2 cells compared to the cells treated with negative sequences only (mock). The trastuzumab-resistant SKBR3/100-8 and BT474/100-2 cells showed more invaded cells compared to parental cell lines, SKBR3, and BT474. Trastuzumab treatment did not reduce the number of invaded cells (Figure 2C,D). Furthermore, the number of invaded cells was significantly reduced when treated with siRNA MALAT1 compared to the cells treated with negative sequences (Figure 2C,D). Trastuzumab treatment further reduced the number of invaded cells in the siRNA MALAT1 treated SKBR3/100-8 and BT474/100-2 cells significantly (Figure 2C,D). The data indicate that the expression of MALAT1 in HER2+ cells mediates the response to trastuzumab treatment. The downregulation of MALAT1 enhanced trastuzumab-inhibiting cell viability and reduced the number of invaded cells, suggesting a potential role for MALAT1 in decreasing sensitivity of trastuzumab in HER2+ breast cancer cells.

### 2.3. Downregulation of MALAT1 Inhibits Epithelial to Mesenchymal Transition-Like (EMT) Phenotype of Breast Cancer Cells

In our previous study, we showed that the activation of the Wnt3/β-catenin signaling pathway promotes EMT-like transition with an increase in cell invasion. These changes were critical mechanisms that led to trastuzumab-resistance in the SKBR3/100-8 and BT474/100-2 cells [44]. The data in Figure 3A,B show that both SKBR3/100-8 and BT474/100-2 had a significant increase in EMT markers, Snail, Slug, Twist, and Nanog, compared to their parental cell lines. Downregulation of Snail, Slug, Twist, and Nanog observed in SKBR3 after knockdown of MALAT1 (Appendix A). The knockdown of MALAT1 in the resistant cell lines also significantly downregulated those EMT markers (Figure 3B). Next, SKBR3 and BT474 cells treated with recombinant Wnt3 protein confirmed the inhibition of cell invasion by knockdown of MALAT1. The data showed that Wnt3 protein treatment significantly increased the number of invaded cells in SKBR3 and BT474 cells, knockdown of MALAT1 inhibited the Wnt3 protein-induced cell invasion (Figure 3C). Unlike the trastuzumab-resistant cells, we observed significantly reduced invaded cells by MALAT1 knockdown in the trastuzumab-sensitive cells, SKBR3 and BT474 (Figure 3C). The data in Figure 3 indicates a regulatory role for MALAT1 in the Wnt3/β-catenin pathway in HER2+ cells. Downregulation of EMT-like transition and inhibition of cell invasion was also observed in TNBC cells, MB231, upon MALAT1 knockdown (Figure 3D), suggesting knockdown of MALAT1 inhibits EMT in different subtypes of breast cancer as well.

### 2.4. Upregulation of MALAT1 in Breast Cancer is Associated with Akt Activation

Our previous studies also showed that activation of the PI3K/Akt pathway in HER2+ breast cancer cells eliminated the inhibiting role of trastuzumab on cell growth [43]. Compared to parental cell lines SKBR3 and BT474, the trastuzumab-resistant lines, SKBR3/100-8 and BT474/100-2, also showed increased phosphorylated Akt (pAkt) (Figure 4A). To further evaluate the association between Akt activation and upregulation of MALAT1, we used myr-Akt-transfected SKBR3 (SKBR3/AA) and NIH3T3 (NIH3T3/AA) cell lines, to compare with their respective vector-transfected cell lines, SKBR3/V and NIH3T3/V. These cell lines were generated from our previous study [43]. The myr-Akt transfected cells showed an increase in phosphorylation of Akt (pAkt) but had no change in total AKT compared to their vector-transfected cells (Figure 4B-left). The SKBR3/AA cells showed no response to trastuzumab [43]. The data in Figure 4B-right show that MALAT1 expression is significantly upregulated in SKBR3/AA and NIH3T3/AA cells compared to vector-transfected SKBR3/V and NIH3T3/V cells. Next, the activation of Akt-mediated expression of MALAT1 was verified by treating cells with GDC-0941 (pictilisib), a potent inhibitor of PI3Kα/δ, and MK-2206, a highly selective Pan Akt inhibitor. The cell lines, JIMT, MB231, and SKBR3/AA, which express high levels of MALAT1, were treated with 10nM MK-2206 or GDC-0941 for 48 h showed significant downregulation of MALAT1 (Figure 4C). Similarly, the trastuzumab-resistant cells SKBR3/100-8 and BT474, when treated with MK-2206, showed downregulation of MALAT1 (Figure 4D). The data suggest that the PI3K/Akt pathway mediates the expression of MALAT1.

### 2.5. Expression of MALAT1 is Regulated by Transcriptional Factor FOXO1

Our previous study showed that activation of Akt resulted in the dysregulation of FOXO1, which then led to HER2+ breast cancer cells becoming resistant to trastuzumab [43]. A lower nuclear expression level of FOXO1 was seen in the trastuzumab-resistant cells SKBR3/100-8 and BT474/1002 compared to their parental cells as well (Figure 5A-left). Inhibition of Akt by MK-2206 upregulated FOXO1 in SKBR3/100-8 (Figure 5A-right), indicating the PI3K/Akt pathway mediates FOXO1 expression. FOXO1 has been reported to negatively regulate MALAT1 by binding to its promoter in osteosarcoma cells [30]. The transcription factor associated with FOXO1 binding sites on the MALAT1 gene promoter has been identified on chr11: 65255303-65255313 with the binding sequence: TCCTGTTTATG by QIAGEN. To determine the molecular mechanism behind MALAT1 regulation, we examined if FOXO1 can bind to the MALAT1 promoter in breast cancer cells. We found that the occupancy of the MALAT1 promoter by FOXO1 in SKBR3/100-8 cells was reduced compared to SKBR3 cells (Figure 5B).

The data indicates that FOXO1 interacts with the promoter of MALAT1. To further investigate the suppressing role of FOXO1 on the MALAT1 promoter, we stably transfected full length human FOXO1 gene or empty vector pCDNA3 into MB231, which is a low FOXO1 expressing cell line. An inverse correlation was seen between MALAT1 expression and FOXO1 expression. As shown in Figure 5A, the expression of FOXO1 increased by 12-fold and 8-fold in the FOXO1 transfected lines, MB231-FOXO1-4 and MB231-FOXO1-18, compared to pCDNA3 transfected cells, respectively (Figure 5C-right). Correspondingly, the expression of MALAT1 had about 2-fold decrease in MB231-FOXO1-4 and MB231-FOXO1-18 cells, compared to MB231-pCDNA3 cells (Figure 5C-left). The MB231-FOXO1-4 cells showed a 4-fold increase in occupancy of the MALAT1 promoter compared to MB231-pCDNA3 cells (Figure 5D). This data indicates that interaction between FOXO1 and MALAT1 promoter occurs in breast cancer cells. Our finding suggests a possible mechanism for MALAT1 in inducing EMT and decreasing the sensitivity of trastuzumab in HER2+ cells via PI3K/Akt and FOXO1. The nuclear export of FOXO1 by elevated pAkt and leading to loss of control MALAT1 at its promoter might be a mechanism for MALAT1 mediated resistance and cell invasion in HER2+ breast cancer cells.

### 2.6. Overexpressing FOXO1 Downregulates MALAT1 and Inhibits Tumor Formation In Vivo

As shown in Figure 6A, the MB231 cells overexpressing FOXO1 had increased expression of E-cadherin and decreased expression of Snail, Slug, Twist, Nanog, and CD44. After MB231-FOXO1-4 and MB231-FOXO-18 cells were injected into nude mice, a significant reduction of tumor growth was observed compared to the mice injected with MB231-pCDNA3 cells (Figure 6B,C). Figure 6B shows examples of tumors’ size between mice injected with MB231-pCDNA3 and MB231-FOXO1-4. We observed tumor growth after one week in mice injected with MB231-pCDNA3 cells, and the dimensions of tumors from MB231-pCDNA3 injection had reached the size limit (according to IACUC guideline) at week 6 (Figure 6C). Furthermore, the tumor growth was much slower in mice injected with MB231-FOXO1-4 and MB231-FOXO-18 cells compared to the mice injected with MB231-pCDNA3 cells. Fewer tumors were observed in those mice injected with MB231-FOXO1-4 or MB231-FOXO1-18 cells at week 3, and the sizes of tumors were much smaller compared to the mice injected with MB231-pCDNA3 (Figure 6C). Compared to mice injected with MB231-FOXO1-4, the size of tumors was larger in mice injected with MB231-FOXO1-18 cells. In the end, MALAT1 expression in those tumor tissues was determined. As shown in Figure 6D, MALAT1 levels were significantly lower in tumors resulting from MB231-FOXO1-4 and MB231-FOXO1-18 cell injection compared to tumors resulting from MB231-pCDNA3 cell injection, suggesting that FOXO1 mediates MALAT1 expression and mammary tumor growth in mice.

## 3. Discussion

An increasing number of studies have documented the clinical significance of MALAT1 in predicting cancer progression [11,12,13,14,15,16,17,18,19,20,21,22,23], and in playing diverse roles in regulating gene transcription, post-transcription, translation, and epigenetic modification [45,46,47].

The oncogenic role of MALAT1 has been reported in various cancers, including lung, colon, cervical, ovarian, pancreatic, bladder, and prostate cancers [11,12,13,14,15,16,17,18], and has been associated with tumor metastasis [11,13,14,15,16,18]. Elevated expression of MALAT1 has been shown in breast cancer tissues and was associated with lymph node metastasis and poor disease outcome as well [19,20,21,23].

Evidence from clinical studies for a role for MALAT1 in breast cancers has been accumulating. Zidan et al., found that MALAT1 expression was significantly elevated in breast cancer cases compared to controls [19]. In their study comprised of 80 patients with breast cancer and 80 controls, MALAT1 expression was shown to be positively correlated with lymph node, ER status, tumor stage, and histological grade, indicating its possible prognostic value [19]. Ou et al., using TNBC tissue microarrays of 240 patients, also found that high MALAT1 expression was significantly correlated with positive lymph node, tumor stage, distant metastasis, and shorter overall survival (OS) and disease-free survival (DFS) of patients [20]. Results consistent with these findings were reported by Miao et al. [21]. They examined MALAT1 expression in 78 breast cancer patients and observed significant upregulation of MALAT1 in cancerous tissues compared with the paired no-cancerous tissues, and an association of high MALAT1 expression with lymph metastasis and shorter DFS in breast cancer tissues [21]. In the same study, Miao et al., investigated the role of MALAT1 in knockdown experiments in cell lines as well. Using lentivirus-mediated RNA interference of MALAT1, they showed that suppression of MALAT1 significantly inhibited MDA-MB231 cell proliferation, migration and invasion, induced apoptosis, and cell cycle G1 arrest [21]. The association of high MALAT1 and shorter DFS in breast cancer was also reported by Wang et al. using the combined data from their study, the 8 Gene Expression Omnibus (GEO) datasets plus TCGA breast cancer provisional data [22].

Similarly, using RNA-seq data and the clinical data from 1086 patients in the TCGA breast cancer cohort at cBioportal, Zheng et al. found a significant negative correlation between OS and the expression of MALAT1 in patients diagnosed at age below 60 or in patients with infiltrating ductal carcinoma [23]. Sun et al., showed in their study that the expression of MALAT1 was up-regulated in primary breast cancer and down-regulated in patients treated with breast-conserving surgery combined with neoadjuvant chemotherapy [24]. Expression of MALAT1 in breast cancers has the potential to predict the response to cancer treatment and cancer prognosis. Kim’s study also analyzed the RNA-Seq data from TCGA but reported contradictory findings that MALAT1 was significantly downregulated in human breast tumors compared with normal mammary tissues [26].

The results from our study bolster the findings from most studies and concur with an oncogenic role of MALAT1 in breast cancer. We show elevated MALAT1 in breast cancer tissues compared to non-cancerous tissues. MALAT1 expression in breast cancer was associated with lymph metastasis. Since we evaluated a limited number of breast cancer tissues, we are unable to determine its association with the different subtypes of breast cancer. However, our in vitro cell model shows upregulation of MALAT1 in all subtypes of breast cancer cells compared to non-cancer cells. We evaluated the expression of MALAT1 in seven breast cancer cell lines that included ER+/HER2-, ER+/HER2+, ER-/HER2+, and TNBC cell lines. Of the different subtypes of breast cancer cell lines, the highest level of MALAT1 was seen in metastatic TNBC and trastuzumab-resistant HER2+ cells.

Different conclusions from two knockdown studies confound the role of MALAT1 in breast cancer pathogenesis. In vivo studies conducted by Arun et al., confirmed the functional role of MALAT1 in regulating critical processes in mammary cancer pathogenesis [25]. Arun et al., used antisense oligonucleotides (ASOs) to knock down MALAT1 in an MMTV (mouse mammary tumor virus)-PyMT mouse mammary carcinoma model which results in slower tumor growth and reduction of metastasis [25]. Furthermore, knockdown of MALAT1 results in a decrease of branching morphogenesis in MMTV-PyMT- and Her2/neu-amplified tumor organoids, increased cell adhesion, and loss of migration [25]. However, Kim’s study reported an opposite phenotype upon loss of MALAT1 in the same mouse background MMTV-PyMT model [26]. Kim et al., used the CRISPR-Cas9-based MALAT1 knockout strategy and observed an increase in metastasis [26]. Furthermore, Kim et al., showed that overexpression of full-length MALAT1 suppressed breast cancer metastasis in the transgenic, xenograft, and syngeneic models [26]. It is unclear if the two different approaches in developing the MALAT1 knockdown mice contribute to the discrepancy in the outcome.

Although the knockdown mice models from both the Arun and Kim studies did not appear to show phenotypic abnormalities, it is essential to note that the ASOs used in Arun’s study lacked a 1.3 kb region upstream of the MALAT1 transcription start site (TSS), the TSS, and a 1.7 kb region downstream of the TSS [48]. The knockdown mice in Kim’s study retained the MALAT1 promoter, including the TSS, and contained a lacZ construct with a polyA tail inserted 69 nt downstream of the MALAT1 TSS [49]. These different observations on the role of MALAT1 in breast cancer warrant additional studies to explore the functional role of MALAT1.

Furthermore, MALAT1-mediated tumor metastasis has been associated with chemoresistance in gastric, lung, cervical, ovarian, and colorectal cancers, and the underlying mechanisms were associated with different cell signaling pathways [27,28,29,50,51,52]. High expression of MALAT1 decreased cisplatin sensitivity in lung cancer, and the mechanism is associated with upregulated Multidrug resistance-associated protein 1 (MRP1) and Multi-Drug resistance 1 (MDR1) via STAT3 [50]. MALAT1 also has been reported interaction with miR-218 decreasing E-cadherin and promoting oxaliplatin-based chemotherapy resistance in colorectal cancer through EZH2-mediating H3K27-me3 [52]. The PIK3/Akt pathway has been shown to play a critical role in MALAT1 induced metastasis and cisplatin resistance in gastric, cervical, and ovarian cancers [27,28,29]. In addition, MALAT1 decreased sensitivity of cisplatin in ovarian cancer was also reported to be associated with the Notch1 signaling pathway [51].

We showed in this study that the upregulation of MALAT1 decreased trastuzumab resistivity in HER2+ breast cancer cells. The PI3K/Akt pathway mediating FOXO1 binding to the promoter of MALAT1 could be a mechanism for MALAT1 inducing EMT and reducing trastuzumab sensitivity in HER2+ breast cancer. FOXO1 was also reported to negatively regulate MALAT1 by binding to its promoter in osteosarcoma cells [30]. Activation of the PI3K/Akt pathway has been seen in many cancer cells and associated with drug resistance [32,33,34]. Expression of myr-Akt in mammary glands accelerates carcinogen-induced tumorigenesis in transgenic mice [53]. FOXO1 can act as a tumor suppressor in several cancers, including breast cancer [38,39,40]. However, the activation of PI3K and phosphorylation of Akt leads to phosphorylation of FOXO1 at different sites. Stimulating its nuclear exit results in the inactivation of FOXO1 [41]. In vivo study from Sinha et al. showed high levels of Cep55 resulted in phosphorylation of Akt protein at ser473 in transgenic mice testis and suppression of FOXO1 protein nuclear retention [37]. We also showed in our earlier study that phosphorylation of Akt protein at ser473 mediates nuclear export of FOXO1 and loss of p27^kip1^ reduced sensitivity of trastuzumab in HER2+ breast cancer cells [43]. We found in this study that overexpressing FOXO1 in metastatic MB231 cells enhanced the occupancy of FOXO1 in the promoter of MALAT1. Inhibition of PI3K/Akt pathway by its inhibitors, MK-2066, and GDC0941, revealed the upregulation of FOXO1 and downregulation of MALAT1 in breast cancer cells. Overexpressing FOXO1 reversed the EMT-like phenotype in vitro and inhibited tumor growth in vivo. We show that overexpression of FOXO1 in tumor tissues from xenograft mouse decreased MALAT1 expression significantly.

Furthermore, we investigated the role of MALAT1 in EMT. The knockdown of MALAT1 in vitro showed inhibition of EMT by decreased expression of Snail, Slug, Twist, and Nanog. We also showed inhibition of cell invasiveness in this study. Activation of the Wnt3/β-catenin signaling pathway promoting EMT-like transition has been identified to be an important mechanism leading to SKBR3 and BT474 cells resistance to trastuzumab in our previous study [44]. We show in this study that the knockdown of MALAT1 reduced the number of invaded cells induced by exogenous Wnt3 protein in HER2+ breast cancer cells, which suggests that MALAT1 may also interact with Wnt3 to induce cell invasiveness and decrease trastuzumab sensitivity in HER2+ breast cancer cells.

There are some limitations to this study. Due to the intra-tumor heterogeneity of breast cancer, assessing cell invasion in vitro, as evaluated by the Boyden Chamber Invasion assay together with examining the expression of EMT markers, may not be sufficient to show inhibition conclusively [54]. Similarly, MTT assay is more likely to measure cell viability based on the absorbance of the resulting formazan solution [55]. An in vitro study using the DNA synthesis method will help to assess the role of MALAT1 in cell proliferation. Further investigation is warranted using more biomarkers associated with tumor metastasis and designing a cohort study with a larger sample size of breast cancer patients to assess the role of MALAT1 in breast cancer invasion, metastasis, and disease outcome.

Trastuzumab is a very effective inhibitor for most, but not all, HER2 positive breast cancers. Clinical studies show that about 52% of HER2 positive breast cancer patients may become resistant to trastuzumab treatment, resulting in breast cancer metastasis and poor survival [2,3]. Our current study suggests that MALAT1 could be a biomarker for predicting response to trastuzumab in HER2+ breast cancer. Therapeutic targeting of the PI3K/Akt pathway and nuclear retention of FOXO1 could suppress the upregulation of MALAT1, re-sensitize the sensitivity of trastuzumab, and prevent breast cancer progression.

In summary, the data from our study support the oncogenic role of MALAT1 in breast cancer. Upregulation of MALAT1 in breast cancer induces EMT and decreases trastuzumab sensitivity in HER2+ breast cancer, and the mechanism is associated with PI3/Akt-mediated FOXO1 nuclear retention. MALAT1 is a potential biomarker for predicting trastuzumab resistance in HER2+ breast cancer.

## 4. Materials and Methods

### 4.1. Cell Cultures, Antibodies, and Reagents

The human breast cancer cell lines SKBR3 (ER/PR-/HER2+) (ATCC: HTB-30), BT474 (ER/PR+/HER2+) (ATCC: HTB-20), MCF7 (ER+) (ATCC: HTB-22), T47D (ER+) (ATCC: HTB-133), MDA-MB231 (MB231, TNBC) (ATCC: HTB-26), human non-breast cancer cell line MCF-12A (ATCC: CRL-10782) were obtained from the American Type Culture Collection (Manassas, VA, USA). HER2-positive trastuzumab-resistant cell line JIMT1 (ER/PR-/HER2+) (DSMZ#: ACC 589) was obtained from Leibniz Institute DSMZ-German Collection of Microorganisms and Cell Cultures (Braunschweig, Germany). Unless otherwise stated, monolayer cultures of all breast cancer cell lines were maintained in DMEM/F12 medium (Cat#: 11320033, Thermo Fisher Scientific, Waltham, MA, USA) with 10% fetal bovine serum (Cat#: 10438026, Thermo Fisher Scientific). MCF12A cells were maintained in DMEM/F12 medium containing 20 ng/mL Human epidermal growth factor (Cat#: PHG0313, Thermo Fisher Scientific), 100 ng/mL cholera toxin (Cat#: 227036, Sigma-Aldrich, St. Louis, MO, USA), 0.01 mg/mL bovine insulin (Cat#: 128-100, Cell Applications, Inc. San Diego, CA, USA), 500 ng/mL hydrocortisone (Cat#: 50-23-7, Sigma-Aldrich), and 5% horse serum (Cat#: H1138, Sigma-Aldrich). The trastuzumab-resistant clones, SKBR3/100-8, and BT474/100-2 cell lines were generated from parental SKBR3 and BT474 cells, respectively. These cell lines were grown in a growth medium containing trastuzumab at 100 µg/mL [23]. SKBR3/AA were SKBR3 cells stably transfected with myr-Akt1 [21].

The antibodies against Phospho-Akt (Ser473) (Cat#: 9271), Akt (pan) (C67E7, Cat#: 4691) were obtained from Cell Signaling Technology, Inc. (Danvers, MA, USA) and monoclonal Anti-β-Actin antibody (Cat#: A2228) obtained from Sigma-Aldrich. PI3K inhibitor, GDC-0941 (Pictilisib, Cat#: HY-50094) and pan-Akt inhibitor, MK-2206 (Cat#: HY-10358) obtained from MedChemExpress, Monmouth Junction, NJ, USA. Recombinant Human Wnt-3 Protein (Cat#: 5036-WN) was purchased from R&D Systems, Inc. (Minneapolis, MN, USA).

### 4.2. MALAT1 Expression in Breast Cancer Tissue

Breast cancer tissues and non-cancer tissues were obtained from the breast tissue bank in the Integrated Clinical and Tissue Biobank/Biorepository Core in our cancer center. The majority of tissues in the breast tissue bank were from African American and Hispanic/Latina women living in South Los Angeles in California, and either examined in the Mammography Clinic or treated in the Hematology/Oncology Clinic at the Martin Luther King Ambulatory Care Center in Los Angeles. The clinical information was collected for each patient by the Core. All breast cancer patients have been followed up at least for 5-years in the Core. MALAT1 expression in breast cancer and non-cancer tissues was examined by mRNA-sequencing (mRNA-seq). RNA was extracted from breast cancer and benign tissues using the RNeasy micro kit (Cat#: 74004, QIAGEN, Germantown, MD, USA). Nanodrop and Qubit Fluorometric instruments were used to determine RNA concentrations. The quality of RNA was determined via capillary electrophoresis using the Agilent 2100 Bioanalyzer (Agilent, Santa Clara, CA, USA). One microgram of RNA was used to construct libraries with KAPA mRNA Hyperprep Kit (Cat#: KK8580, Roche Sequencing Solutions, Indianapolis, IN, USA). Libraries were sequenced on a Hiseq3000 instrument (Illumina, San Diego, CA, USA). Single-ended 50 bp mRNA-seq reads were aligned to hg19 using default parameters of Tophat2 (version 2.1.0) and Bowtie2 (version 2.3.2). Samtools (version 0.1.18) was used to convert SAM to BAM files. FPKM values were generated using default parameters for Cufflinks (version 2.1.1). Only FPKM values greater than 0.5 were considered for further analysis. Cufflinks output statistical analysis was utilized to determine significance.

### 4.3. Quantitative Real-Time Reverse Transcription-PCR (RT-qPCR)

RT-qPCR was performed with iCycler iQ Real-Time PCR Detection System (Bio-Rad Laboratories, Hercules, CA, USA) using SYBR Green Master Mix (Cat#: 180820, QIAGEN). The mRNA levels of genes were quantified by measuring the threshold cycle (Ct) and adjusted with the level of 18S for each sample. QIAGEN synthesized primer sequences for MALAT1 shown below:
MALAT1-Forward: 5′-AAAGCAAGGTCTCCCCACAAG-3′.MALAT1-Reverse: 5′-GGTCTGTGCTAGATCAAAAGGCA-3′.


### 4.4. Overexpressing FOXO1 Gene

MB231-FOXO1-4 and MB231-FOXO1-18 were clonal selections generated by stably transfected full-length FOXO1 gene (Addgene plasmid 13507) in MB231 cells, and MB231-pCDNA3 was MB231 stably transfected with pCDNA3 vector only. The transfected cells were maintained in a growth media containing 400 μg/mL G418 (gentamicin) disulfate salt solution (Cat#: G8168, Sigma-Aldrich). Overexpressing FOXO1 in the cells was confirmed by RT-qPCR with primers: FOXO1-Forward: ′5-TTTGGACTGCTTCTCTCAGTTCCTGC-3′ and FOXO1-Reverse: 5′-TTTGACAATGTGTTGCCCAA CCAAAG-3′.

### 4.5. MALAT1 siRNA Transfection

Cells were plated in six-well plates at 2.0 × 10^5^ cells per well for 24 h with the serum-free medium before transfection. MALAT1 siRNA (Cat#: 4390771) and control negative siRNA (Mock, Cat#: 4390843) were obtained from Thermo Fisher Scientific. Lipofectamine™ RNAiMAX transfection reagent (Cat#: 13778075, Thermo Fisher Scientific) was used for transfection following the manufacturer’s instructions. Gene expression after siRNA knockdown was determined by PCR or quantitative reverse transcription-PCR (RT-Q-PCR) after 72 h of transfection.

### 4.6. Boyden Chamber Invasion Assay

The cells were either treated with siRNA MALAT1 or negative sequences for 72 h or after 24 h, then co-treated with human Wnt3 recombinant protein at 50 ng/mL for an additional 48 h. The invasive assay was done in 24-well cell culture chambers using inserts with 8 μm pore membranes pre-coated with Matrigel (Cat#: A1569601, Thermo Fisher Scientific). The test cells were placed in the upper wells without serum and conditioned medium from metastatic TNBC cells MB231, were plated in the lower wells. After 24 h, the cells were fixed by 0.5% glutaraldehyde and stained with 0.5% toluidine blue. The numbers of invaded cells were counted utilizing a 20× objective microscope from three files per membrane and then normalized with the total number of cells. Each experiment was performed twice.

### 4.7. Cell Viability Assay

Cells were plated on to 96-well plates, and MALAT siRNA was added to the test wells. After 24 h of adding trastuzumab (gifted by Genentech, San Francisco, CA, USA) into each well at 0, 10 µg/mL, 20 µg/mL, 50 µg and 100 µg/mL for an additional 48 h, cell viability was determined using the MTT (3-(4,5-dimethylthiazol-2-yl)-2,5-diphenyltetrazolium bromide) assay.

### 4.8. Chromatin Immunoprecipitation-Real-Time PCR Assay (ChIP-qPCR)

FOXO1 Binding Sites on MALAT1 gene promoter have been identified on chr11: 65255303-65255313 and binding sequence: TCCTGTTTATG by ChIP assay (QIAGEN). The chromatin-protein complex from the cells was isolated by immunoprecipitation of the chromatin with FOXO1 antibody (ab39670, Abcam, Cambridge, MA, USA) using Magna-ChIP assay kit (Cat# 17-10085, Sigma-Aldrich) following the manufacture’s instruction and then qPCR with MALAT1 promoter primers. The primer covering the FOXO1-binding region of the promoter of MALAT1 was designed and synthesized by QIAGEN. Fold enrichment was calculated as the amount precipitated by anti-FOXO1 IgG relative to the amount precipitated by normal IgG (mock samples). The data were also calculated as % input to verify that results were consistent.

### 4.9. An Animal Study In Vivo

The animal study was conducted after approval by our University’s IACUC. Briefly, MB231-FOXO1 and MB231-pCDN3 cells (1.0 × 10^6^ cells) were injected into the first and the sixth mammary fat pads on the right and left flanks of 4-week female athymic nude mice (Charles River Laboratories, Hollister, CA, USA). The tumor growth was checked twice per week after injection with both manual caliper and Near-infrared fluorescence small animal imaging system (LI-COR Biotechnology, Lincoln, NE, USA) until tumor volume reached the IACUC’s limited protocol size (1.5 cm^3^). The tumor tissue was collected for further analysis.

### 4.10. Statistical Analysis

All experiments were independently conducted four times in total. Each analysis was performed at least twice with four different independent times of sample collections, i.e., RNA was extracted from the cells from four different treatments. The cells used for each MTT and invasion assays were from four different cultures. The experiment conducted from the same samples’ preparation was also conducted at least two to three times. Three determinations for qPCR and invasion assays and 8 determinations for MTT were required at each performance. The data were presented as mean ± SEM from the four independent experiments, and the differences between groups were compared by ANOVA test for mean. The statistical significance was presented as “* *p* < 0.05” if the *p*-value is less than 0.05, or as “** *p* < 0.01” if the *p*-value is less than 0.01. The MALAT1 levels in breast tissue samples were presented as a median level, and the Mann-Whitney U test was used to determine the statistical significance between the groups.

## 5. Conclusions

Data from our study demonstrates the oncogenic role of MALAT1 in breast cancer. We show that MALAT1 contributes to HER2+ cell resistance to trastuzumab. Targeting or inhibiting the PI3/Akt pathway and stabilizing FOXO1 translocation could inhibit the upregulation of MALAT1. This strategy has the potential to overcome trastuzumab resistance in HER2 overexpressing breast cancers.

## Figures and Tables

**Figure 1 cancers-12-01918-f001:**
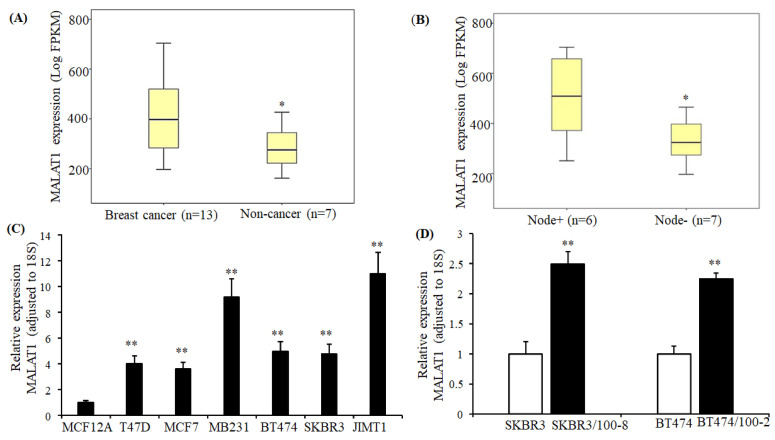
MALAT1 expression in breast cancer tissues and cell lines. (**A**). Tissue samples from breast cancer and non-cancer tissues were examined by mRNA-sequencing, as described in Methods. The box shows the range of expression of MALAT1 in breast cancer and non-cancer tissues and the horizontal line the median level. High MALAT1 expression was detected in breast cancer tissues, * *p* < 0.05 compared to non-cancer tissues. The Mann-Whitney U test determined the significance; (**B**). Tissue samples from breast cancer patients were grouped with and without lymph node involvement. The box shows the range of expression of MALAT1 in breast cancer with positive and negative lymph nodes, and the horizontal line indicates the median level. Breast cancer tissue with positive lymph nodes showed an increased MALAT1 level. * *p* < 0.05 compared to non-cancer tissues. The Mann-Whitney U test determined the significance; (**C**,**D**). Total RNA extracted from the cell lines, analyzed by Quantitative real-time reverse transcription-PCR (RT-qPCR) for expression of MALAT1, and adjusted for 18S. The bar graphs in C and D indicate the related mean ± SEM from four repeated experiments. Breast cancer cell lines showed increased MALAT1 compared to MCF12A. ** *p* < 0.01 comparing the non-cancer cell line (MCF-12A) with the breast tumor lines in Figure **C**, and ** *p* < 0.01 when comparing parental lines, SKBR3 and BT474, to their derivatives in Figure **D**. The ANOVA test determined the significance. Both trastuzumab-resistant cell lines increased MALAT1 compared to their parental lines.

**Figure 2 cancers-12-01918-f002:**
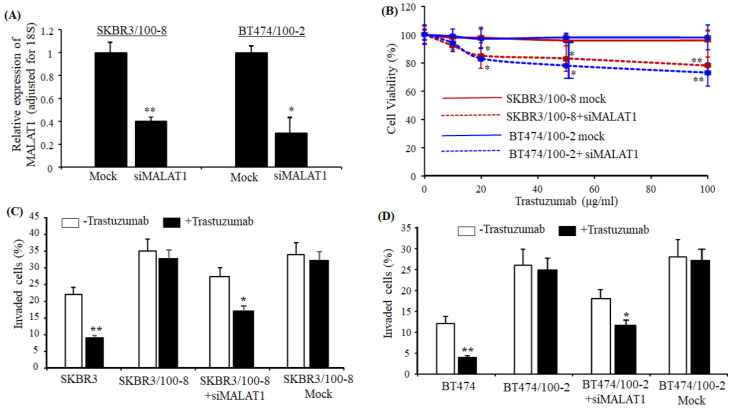
Knockdown of MALAT1 increases HER2-cells sensitivity of trastuzumab. (**A**). SKBR3/100-8 and BT474/100-2 cells were treated with siRNA MALAT1 or negative sequences (mock) for 72 h as described in Methods, and RNA extracted. The bar graphs indicate the relative levels of MALAT1 (mean ± SEM) as determined by RT-qPCR from four repeated experiments and show MALAT1 knockdown cells have downregulated MALAT1, * *p* < 0.05, and ** *p* < 0.01 compared to mock cell lines. The ANOVA test determined the significance; (**B**). SKBR3/100-8 and BT474/100-2 treated with siRNA MALAT1 or negative sequences (mock) for 24 h, were co-treated with trastuzumab at the indicated doses either with siRNA MALAT1 or negative sequences for an additional 48 h. MTT assay determined cell viability. Each data point was from six measurements, and the experiments were independently performed four times. Red color indicates SKBR3/100-8 and blue color indicates BT474. The graph shows the mean ± SEM from four repeated tests. A statistically significant change in cell viability was observed between the siRNA-treated SKBR3/100-8 (red dotted line) and BT474/100-2 (blue dotted line). siRNA treatment decreased cell viability in the trastuzumab-resistant HER2+, * *p* < 0.05, and ** *p* < 0.01 compared to their untreated cells, respectively. The ANOVA test determined the significance; (**C**,**D**). SKBR3/100-8 and BT474/1002 cells treated with siRNA MALAT1 or negative sequence (mock) for 24 h, were co-treated with or without trastuzumab at 20µg/mL for an additional 48 h. The invaded cells were measured by the Boyden Chamber Invasion assay, as described in the Methods section. The mean invaded cells were counted from five different areas first for each time of the experiment. The bar in the figure showed the mean ± SEM from four independent experiments. The number of invading cells was reduced significantly upon combination treatment of siRNA and trastuzumab in the resistant cells, * *p* < 0.05, and ** *p* < 0.01 compared to their respective untreated cells. The ANOVA test determined the significance.

**Figure 3 cancers-12-01918-f003:**
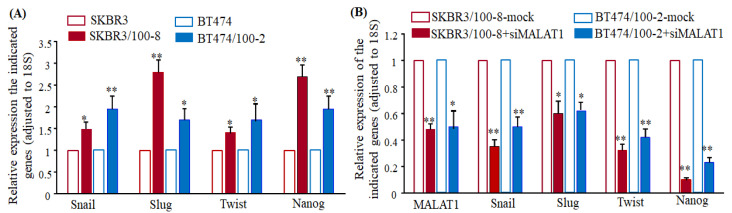
The knockdown of MALAT1 downregulates EMT makers. (**A**,**B**). Total RNA was extracted from the cells, and gene expression was quantified using RT-qPCR. The bar graphs show levels of the indicated genes relative to 18S. Each bar indicates the mean ± SEM from 4 independent experiments. The knockdown of MALAT1 downregulates EMT markers in the trastuzumab-resistant cells, * *p* < 0.05 and ** *p* < 0.01, compared to respective parental cells SKBR3 or BT474, (**A**), or their corresponding mock cell lines (**B**). The ANOVA test was used to determine significance. (**C**). MALAT1 siRNA treatment reduced the number of invaded cells. Cell invasiveness was measured by the Boyden Chamber Invasion assay, as described in the Methods section. The number of invaded cells from the first five random spots was counted. Mean values were calculated from four independent measurements. The bars indicate mean ± SEM. ** *p* < 0.01 observed when SKBR3 cells compared without siRNA MALAT1 treatment and * *p* < 0.01 compared to their respective cell lines without siRNA treatment. The ANOVA test was used to determine significance. (**D**). Left-Panel MB231 cells were treated with siRNA MALAT1 or negative sequences (mock) for 72 h. RT-qPCR was used to analyze the indicated genes. Each bar shows the mean ± SEM from four independent experiments. The data suggest siRNA treatment inhibited EMT markers in TNBC cells (* *p* < 0.05 and ** *p* < 0.01) compared to mock cells. Right Panel: The bar graph shows a reduced number of invaded cells by MALAt1 siRNA treatment as determined by the Boyden Chamber Invasion assay. Each bar indicates the mean ± SEM from four independent experiments. The number of invaded cells from the first five random spots was counted in each analysis. * *p* < 0.05 compared to mock cells. The ANOVA test was used to determine significance.

**Figure 4 cancers-12-01918-f004:**
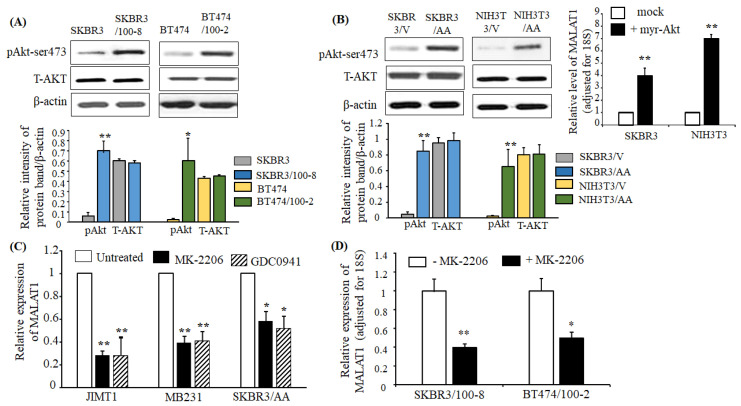
The PI3K/Akt pathway mediates the expression of MALAT1. (**A**) Total protein was extracted from each cell line, and Western blot analysis was performed with antibodies specific to phosphorylated Akt at ser-473 (pAkt) and total AKT (T-AKT). β-actin was used for loading control. The top panel shows representative Western blot images, and the bottom panel shows quantification by densitometric analysis of Western blots from four independent experiments. pAkt and T-AKT protein levels are shown relative to β-actin. The bars indicate mean ± SEM. Increased pAkt was observed in the trastuzumab-resistant cells compared to their parental lines, respectively (* *p* < 0.05 and ** *p* < 0.01). An ANOVA test was used to determine significance. (**B**) SKBR3 and NIH3T3 were stably transfected with myr-Akt (+myr-Akt) or with vector only (mock). The Western blot images in the top-left panel show the representative protein expressions of pAkt, T-AKT, and β-actin. The bottom panel shows quantification by densitometric analysis of protein (pAkt/β-actin or T-AKT/β-actin) levels from four different Western blots. The bars indicate mean ± SEM and demonstrate elevated pAkt in SKBR3/AA and NIH3T3/AA (** *p* < 0.01) compared to the respective SKBR3/V and NIH3T3/V cell lines. An ANOVA test was used to determine significance. The top-right panel shows the levels of MALAT1 relative to 18S RNA, as determined by RT-qPCR. The bar graph indicates the mean ± SEM from four independent experiments. Myr-Akt-transfected cells show increased MALAT1 expression, ** *p* < 0.01, compared to mock cells. Significance was determined using the ANOVA test. (**C**) The indicated cell lines were treated with MK-2206 or GDC0941 for 48 h, and RT-qPCR was used to determine MALAT1 relative to 18S expression. Each bar indicates mean ± SEM from four independent experiments. The data showed that MK-226 and DGC0941 inhibited expression of MALAT1 in JIMT, MB231, and SKBR3/AA cells (* *p* < 0.05 and ** *p* < 0.01) compared with their respective untreated cells. An ANOVA test was used to determine significance. (**D**) MK-2206 inhibited MALAT1 expression in trastuzumab-resistant cells. Expression of MALAT1, as determined by RT-qPCR, in SKBR3/100-8 and BT474 cells were treated with MK-226 for 48 h. The bar graph shows the level of MALAT1 adjusted for 18S. Each bar is the mean ± SEM from four independent experiments, with * *p* < 0.05, ** *p* < 0.01 compared to their respective untreated cells as determined by the ANOVA test.

**Figure 5 cancers-12-01918-f005:**
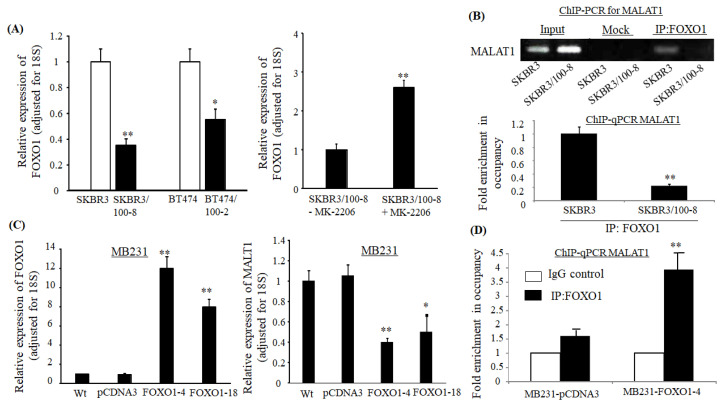
FOXO1 regulates the expression of MALAT1. (**A**). Total RNA extracted from SKBR3, SKBR/100-8, BT474, and BT474/100-2 (left panel), and SKBR3/100-8 cells treated with MK-226 for 48 h (right panel). RT-qPCR was used to determine the expression of FOXO1, and the bar graphs indicate the level of FOXO1 relative to 18S. Each bar represents mean ± SEM from four independent experiments. A low level of FOXO1 was seen in trastuzumab-resistant cells, * *p* < 0.05 and ** *p* < 0.01, compared to SKBR3 or BT474, respectively. MK-2206 treatment of SKBR3/100-8 increased the FOXO1 level, ** *p* < 0.01, compared to SKBR3/100-8 without MK-226 treatment. An ANOVA test was used to determine significance. (**B**). FOXO1 binding to the promoter of MALAT1 in the indicated cells was determined by individual ChIP-PCR (top, *n* = 3) and ChIP-qPCR (bottom, *n* = 3) assays as described in the Method section. The image at the top shows the photo of a gel from a representative ChIP-PCR using MALAT1-specific primers for each condition. The bar graph in the bottom shows data from ChIP-qPCR of the specified samples. The bars in the graph indicate the fold enrichment (mean ± SEM) of FOXO1 binding to the promoter of MALAT1 in SKBR3 and SKBR3/100-8 cells compared to mock cells (immunoprecipitated with IgG) ** *p* < 0.01 compared to SKBR3 cells as determined by the ANOVA test. (**C**). Total RNA extracted from MB231 (wt), MB231 transfected with vector (pCDNA3), and MB231 stably transfected with full-length FOXO1 (FOXO1-4 and FOXO1-18) cells. RT-qPCR was used to determine the expression of FOXO1 and MALAT1. The bar graphs indicate levels of FOXO1 (left panel) and MALAT1 (right panel) relative to 18S. Each bar shows mean ± SEM from four repeated experiments. Overexpressing FOXO1 in MB231 cells reduced MALAT expression, * *p* < 0.05, and ** *p* < 0.01, compared to wt and pCDNA3 cells as determined by the ANOVA test. (**D**). Individual ChIP-qPCR (*n* = 3) assays were conducted. The bar graph indicates fold enrichment of FOXO1 (mean ± SEM) binding to the promoter of MALAT1 in MB231-pCDNA3 and MB231-FOXO1-4 cells. Overexpression of FOXO1 increased its occupancy of MALAT1 promoter, ** *p* < 0.01, compared to IgG control cells as determined by the ANOVA test.

**Figure 6 cancers-12-01918-f006:**
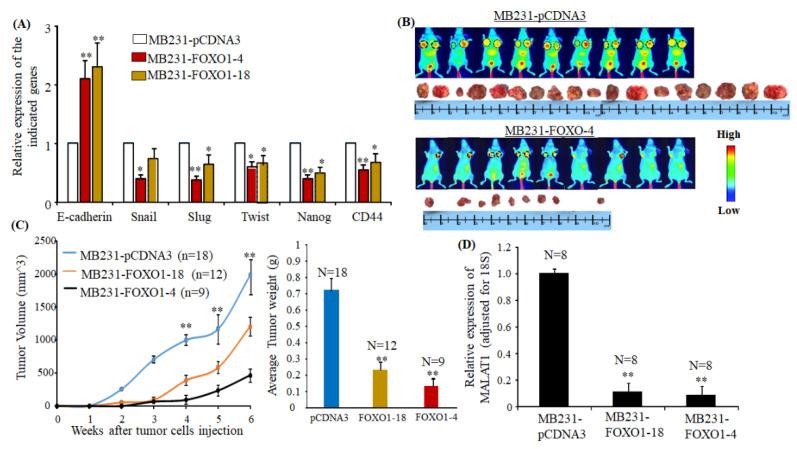
Overexpressing FOXO1 downregulates MALAT1 and inhibits tumor formation in vivo. (**A**) RT-qPCR was used to determine the expression of the indicated genes in MB231-pCDNA3, MB231-FOXO1-4, and MB231-FOXO1-18. The bar graph shows their expression levels adjusted for 18S. Each bar indicates a mean ± SEM from four repeated experiments. FOXO1 transfection reversed EMT marker gene expression, * *p* < 0.05, and ** *p* < 0.01 compared to MB231-pCDNA3 transfection as determined by ANOVA test. (**B**) The images were taken by LI-COR small-animal imaging system before tumors were removed, and the circles designate the tumor images. Representative tumors from the indicated groups are shown in (**B**). Fewer and smaller tumors were observed in the mice injected with MB231-FOXO1-4 cells. (**C**) Left panel: The indicated cells were injected into the mammary fat pads of mice as described in the Methods section. The tumor growths in mice were monitored, and tumor volumes were measured at different times. Each data point in the graph indicates mean ± SEM from the tumors detected in the respective groups. ** *p* < 0.01 compared to the tumor growth by injection with MB231-FOXO1-4 and MB231-FOXO1-18 cells, * *p* < 0.05 compared to the tumor growth by injection with MB231-FOXO1-18 and MB231-FOXO1-4 in the indicated time points. An ANOVA test was used to determine significance. Right panel: the bar graph indicates the mean ± SEM tumor weight from each group. ** *p* < 0.01 compared to the MB231-pCDNA3-cell-injected group as determined by the ANOVA test. (**D**) RNA was extracted from tumors isolated from the three groups, and RT-qPCR was used to determine MALAT1 expression in the tumor tissues. The bar graph indicates levels of MALAT1 relative to 18S. The bars indicate mean ± SEM from experiments repeated three times. MALAT1 levels were significantly lower in tumor tissues from mice injected with MB231-FOXO1-4 and MB231-FOXO1-18 cells compared to mice injected with MB231-pCDN3, ** *p* < 0.01 compared to the tumors from the MB231-pCDNA3-injected group as determined by ANOVA test.

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
