# Peer review of "Expression of MALAT1 Promotes Trastuzumab Resistance in HER2 Overexpressing Breast Cancers"

_cancers, 2020, doi:10.3390/cancers12071918_

Round 1

Reviewer 1 Report

Based on Author’s previous studies, which showed MALAT1 as one of the most significantly upregulated molecules in breast cancer cells resistant to trastuzumab and/or tamoxifen, in this manuscript the Authors examined the role of MALAT1 in breast cancer and its association with HER2+ breast cancer cells resistant to trastuzumab. They demonstrated a molecular mechanism involving PI3/Akt pathway and FOXO1 translocation in the regulation of MALAT1 expression, and suggest that targeting PI3/Akt pathway and stabilizing FOXO1 translocation could inhibit the upregulation of MALAT1 and overcome trastuzumab resistance in HER2 overexpressing breast cancers.

Comments and suggestions:

All statements should be supported by references, like the phase below. 

 “Therapeutic failure and distant metastasis have been significant challenges in the treatment of breast cancer as well as the leading cause of mortality in breast cancer patients. Compared to all different types of breast cancers, HER2 (human epidermal growth factor receptor 2) overexpressing (HER2+) and triple-negative breast cancers (TNBC, estrogen/progesterone receptors negative (ER/PR-) and HER2 -) are more likely to develop the metastatic disease due to their aggressive tumor characteristics”.

  • The Introduction section should be updated with more recent references about MALAT1 and breast cancer, since there are many works in literature and the topic is very much covered.
  • The non-cancer tissues utilized in the RNASeq experiments are matched with cancer tissues, if not explain why not-matched tissues were analyzed.
  • All Figures need of major resolution.
  • Add the letters A and B in the figure 6
  • The discussion section needs a wide revision. The Authors should discuss the literature’s data of their subject, above all the recently published papers with the same subject. Discuss the differences between their work and controversial data published on Kim et al. Nature Genetics 2018.

Author Response

We wish to thank reviewer-1 for the helpful and constructive suggestions. Here are our responses to each reviewer-1 comments/suggestions.

Review 1

Comments and Suggestions for Authors

Based on the Author's previous studies, which showed MALAT1 as one of the most significantly upregulated molecules in breast cancer cells resistant to trastuzumab and/or tamoxifen, in this manuscript the Authors examined the role of MALAT1 in breast cancer and its association with HER2+ breast cancer cells resistant to trastuzumab. They demonstrated a molecular mechanism involving PI3/Akt pathway and FOXO1 translocation in the regulation of MALAT1 expression, and suggest that targeting PI3/Akt pathway and stabilizing FOXO1 translocation could inhibit the upregulation of MALAT1 and overcome trastuzumab resistance in HER2 overexpressing breast cancers.

Comments and suggestions:

All statements should be supported by references, like the phrase below.

 "Therapeutic failure and distant metastasis have been significant challenges in the treatment of breast cancer as well as the leading cause of mortality in breast cancer patients. Compared to all different types of breast cancers, HER2 (human epidermal growth factor receptor 2) overexpressing (HER2+) and triple-negative breast cancers (TNBC, estrogen/progesterone receptors negative (ER/PR-) and HER2 -) are more likely to develop the metastatic disease due to their aggressive tumor characteristics".

Response: As recommended, the references 1-4 have been added to support the above statements (lines: 45).

The Introduction section should be updated with more recent references about MALAT1 and breast cancer since there are many works in literature, and the topic is very much covered.

Response: We have included more recent publications from 2019-2020 (References 20, 23, 24, 28, 31) that have been added in the introduction.

The non-cancer tissues utilized in the RNASeq experiments are matched with cancer tissues, if not explain why not-matched tissues were analyzed.

Response Yes, we did match cancer tissues with non-cancer tissues from the same patient. We have revised the sentence in the revised manuscript as “The non-cancer tissue was the normal tissue adjacent to breast carcinoma from the same patient” (lines: 118-119).

All Figures need of major resolution.

Response: We have improved the resolution of figures and sent high-resolution figures as Zip files separately to the Journal

Add the letters A and B in figure 6

Response: The letters A and B have been added in figure 6.

The discussion section needs a wide revision. The Authors should discuss the literature's data of their subject, above all the recently published papers with the same subject. Discuss the differences between their work and controversial data published on Kim et al. Nature Genetics 2018.

Response: We have made significant revisions in our discussion section, as recommended by the reviewers (highlighted in yellow). We have discussed the literature's data extensively and, in particular, the recent publication by Kim et al. Nature Genetics, 2018. Please review the track changes in the discussion section (lines: 411-441).

Reviewer 2 Report

Comments to the Author

This manuscript has potential and focuses a concerning issue in breast cancer therapy, by elucidating a new molecular mechanism underlying resistance to trastuzumab. The authors reported the main role of MALAT1 upregulation in resistance to trastuzumab. This upregulation is modulated by the PI3K/AKT pathway, through the FOXO1 transcritional factor. The study is relevant; however, it has considerable weaknesses and limitations that need to be clarified. Moreover, the discussion section should be improved, as the results are not discussed adequately and there is a lack of explanation of the mechanisms underlying the results, in order to highlight the importance of this study. In addition, the English should be improved, since there are several grammatical errors. It is strongly recommended to revise this manuscript, prior to any publication. Major revisions are mandatory.

Major comments:

  • In pag3, line 101 the authors describe that “MALAT1 upregulation in breast cancer is associated with HER2+ cells that are resistant to trastuzumab, and with metastatic TNBC” but this is not totally true! After careful analysis of Figure 1, it is possible to conclude that MALAT1 is generally overexpressed in all breast cancer subtypes (ER+, ER+/PR+, TNBC, HER2+/PR+; HER2+ and trastuzumab-resistant HER2+ cells (JIMT1)). In fact, the relative expression levels of MALAT1 are higher in MB231 and JIMT1, when compared with the other cancer cell models, nevertheless it cannot be discarded that this protein is also overexpressed in the other breast cancer cell models, when comparing with the non-breast cancer cell model MCF12A. So the authors should re-phrase the discussion of the results of the Figure 1. Moreover, based on the author’s comments, why do the authors select the SKBR3 and BT474 cell models for the subsequent studies, if these are not the cell models that overexpress MALAT1?

  • In figure 2, the authors demonstrated that knock-down of MALAT1 increases sensitivity to transtuzumab, by impairing cell growth and cell invasion. However, this conclusion, in view of the results presented, has considerable limitations. To study cell growth the authors only performed MTT assay and to study cell invasion the authors only carried out Boyden Chamber Invasion assay. It is too ambitious to extrapolate results based on just one type of method/assay, which is not specific to the mechanism under study. For example, it should be pointed that MTT assay did not study cell growth but only cell metabolic activity or loss of viable cells. To study cell growth, DNA synthesis methods should be performed. Furthermore, to confirm cell migration, beyond Boyden Chamber Invasion assay, another type of  assays, as zymographic analysis or expression levels of biomarkers associated to cell invasion, should be performed, due to the limitations of the Boyden Chamber Invasion assay (Z.I. Stryker, Biomedicines 2019, doi:10.3390/biomedicines7020037; T.S. Gerashchenko, J. Clin. Med. 2019, 1092; doi:10.3390/jcm8081092). 

  • To confirm that MALAT1 is linked to transtuzumab resistance and to confirm that knock-down of MALAT1 increases sensitivity to transtuzumab, the authors should study the effects of MALAT1 knock-down on cell growth, cell invasion and EMT markers in sensitive and resistant transtuzumab cell models. Besides SKBR3/100-8 and BT474/100-2 cells, the authors should applied MALAT1 knock-down on SKBR3 and BT474 cell models and compare the results, in order to confirm that the MALAT1 knock-down effects were different or the same between sensitive and resistant cells and/or if the observed effects were specific of transtuzumab-resistance.

  • The authors demonstrate in Figure 4 the protein expression levels of pAKT Ser473, T-AKT and β-actin. Although, the authors did not say on figure legend the number of experiments performed. So densitometric analysis of at least three independent experiments should be performed and presented.

  • Figure 4: Why the authors studied the relative expression of MALAT1 with or without MK-2206 and GDC0941 in JIMT1, MB231 and SKB3/AA cell models? Why the authors did not performed this assays in SKBR3/100-8, BT474/100-2, SKBR3 and BT474 cell models, if these cells correspond to sensitive and resistant transtuzumab cell models, in order to conclude that MALAT1 upregulation is mediated by PI3K/AKT pathway, being this mechanism linked to transtuzumab-resistance. Explanations of the selection of the cell lines should be performed.

  • In page 7, line 228, the authors say that “Inhibition of Akt by MK-2206 unregulated FOXO1 in SKBR3/100-8” but in BT474/100-2 cell model, what is the effect? Why the authors only used MK-2206 and did not applied GDC0941?

  • Why the authors selected the MB231 cell model for the in vivo studies and not used the other cell models generally applied in the previous in vitro assays? A justification should be provided.

  • Based on author’s data, in discussion section, the authors should discuss the mechanisms by which targeting MALAT1 could overcome HER2+ breast cancer resistance to trastuzumab.

Minor comments:

  • In figure 2B, y-legend, instead of Cell growth (%) the authors should write Cell viability (%), since MTT assay measures the reduction of a tetrazolium component (MTT) into an insoluble formazan product by the mitochondria of viable cells. Thus, MTT assay did not study cell growth but only cell metabolic activity or loss of viable cells.

Author Response

We wish to thank reviewer 2 for the helpful and constructive suggestions. Here are our responses to each reviewer-2 comments/suggestions.

Review 2

Comments and Suggestions for Authors

Comments to the Author

This manuscript has potential and focuses a concerning issue in breast cancer therapy, by elucidating a new molecular mechanism underlying resistance to trastuzumab. The authors reported the main role of MALAT1 upregulation in resistance to trastuzumab. This upregulation is modulated by the PI3K/AKT pathway, through the FOXO1 transcriptional factor. The study is relevant; however, it has considerable weaknesses and limitations that need to be clarified. Moreover, the discussion section should be improved, as the results are not discussed adequately, and there is a lack of explanation of the mechanisms underlying the results in order to highlight the importance of this study. In addition, the English should be improved, since there are several grammatical errors. It is strongly recommended to revise this manuscript, prior to any publication. Major revisions are mandatory.

Response: We wish to thank the reviewer's encouragement. We have made significant revisions in the manuscript according to the reviewer's comments (heighted in yellow in the revision).

Major comments:

On page3, line 101 the authors describe that "MALAT1 upregulation in breast cancer is associated with HER2+ cells that are resistant to trastuzumab, and with metastatic TNBC" but this is not totally true! After careful analysis of Figure 1, it is possible to conclude that MALAT1 is generally overexpressed in all breast cancer subtypes (ER+, ER+/PR+, TNBC, HER2+/PR+; HER2+ and trastuzumab-resistant HER2+ cells (JIMT1)). In fact, the relative expression levels of MALAT1 are higher in MB231 and JIMT1, when compared with the other cancer cell models, nevertheless it cannot be discarded that this protein is also overexpressed in the other breast cancer cell models, when comparing with the non-breast cancer cell model MCF12A. So the authors should re-phrase the discussion of the results of Figure 1.

Response: Yes, we agree with the comments from the reviewer that Data in figure 1 shows that MALAT1 is also upregulated in other breast cancer cell subtypes, compared to non-cancer breast cells, MCF12A. However, when compared with breast cancer subtypes, MALAT1 is mostly elevated in MDA-MA231 (TNBC cells) and JIMT1 (trastuzumab-resistant HER2+ cells). The expression of MALAT1 in ER+/HER2- cell lines such as MCF-7 and T47D are similar, and slightly increased in ER+/HER2+ cells BT474 and ER-/HER2+ cells SKBR3. Based on these data, we concluded that "Compared to non-breast cancer cells MCF12A, expression of MALAT1 was significantly upregulated in all subtypes of breast cancer cells (Figure 1C). The highest expression of MALAT1 was seen in metastatic TNBC cells, MB231, and trastuzumab-resistant HER2+ cells, JIMT1." (Lines: 122-125) in our original version of the manuscript.  To better clarify our findings, we have re-phrased the discussion and conclusion sections related to the data from Figure 1, according to the reviewer's comments, in the revised manuscript as: "The data in Figure 1 suggests that MALAT1 is upregulated in all subtypes of breast cancer. However, the upregulation of MALAT1 is more significantly associated with HER2+ cells that are resistant to trastuzumab, and with metastatic TNBC in breast cancer cells." (Lines 128-131). In the discussion section, we re-phrased our findings as "We show elevated MALAT1 in breast cancer tissues compared to non-cancerous tissues. MALAT1 expression in breast cancer was associated with lymph metastasis. Since we evaluated a limited number of breast cancer tissues, we are unable to determine its association with the different subtypes of breast cancer. However, our in vitro cell model shows upregulation of MALAT1 in all subtypes of breast cancer cells compared to non-cancer cells. We evaluated the expression of MALAT1 in 7 breast cancer cell lines that included ER+/HER2-, ER+/HER2+, ER-/HER2+, and TNBC cell lines. Of the different subtypes of breast cancer cell lines, the highest level of MALAT1 was seen in metastatic TNBC and trastuzumab-resistant HER2+ cells.” (lines 403-410).  

Moreover, based on the Author's comments, why do the authors select the SKBR3 and BT474 cell models for the subsequent studies, if these are not the cell models that overexpress MALAT1?

Response: As we mentioned in the introduction that this study is a continuation of our ongoing studies where we have been interested in elucidating the mechanisms of drug resistance in breast cancer, as well as in other cancers. Understanding trastuzumab resistance in HER2+ cells is one of our primary focuses. To do so, we generated trastuzumab-resistant HER2+ breast cancer cells, SKBR3/100-8 and BT474/100-2, from the parental HER2+ cell lines, SKBR3, and BT474. The resistant cells have been validated to be resistant to trastuzumab and display EMT phenotype (reference 44) in our previous study. Through gene array analysis, and comparing the profiles from resistant cells with sensitive cells (parental lines) with or without trastuzumab treatment, we identified that activation of Wnt3 and β-catenin signaling pathway contributes to trastuzumab resistance significantly (reference 44). Later we also determined that TGFβ/Smad3 signaling can regulate Twist while sustaining the activation of Wnt3/β-catenin (Breast Cancer Res Treat. 2017, 163(3):449-460. PMCID: PMC5425246). In the same gene array analysis, we identified that MALAT1 is the highest upregulated gene in SKBR3/100-8 cells. In this study, we confirmed that MALAT1 is significantly increased in both SKBR3/100-8 and BT474/100-2 trastuzumab-resistant cell lines compared to their respective parental cell lines, SKBR3 and BT474 (Figure 1D). Hence this was the primary reason why we used those parental cell lines as models for this study.

In figure 2, the authors demonstrated that knockdown of MALAT1 increases sensitivity to trastuzumab, by impairing cell growth and cell invasion. However, this conclusion, in view of the results presented, has considerable limitations. To study cell growth, the authors only performed MTT assay, and to study cell invasion, the authors only carried out the Boyden Chamber Invasion assay. It is too ambitious to extrapolate results based on just one type of method/assay, which is not specific to the mechanism under study. For example, it should be pointed out that MTT assay did not study cell growth but only cell metabolic activity or loss of viable cells. To study cell growth, DNA synthesis methods should be performed. Furthermore, to confirm cell migration, beyond Boyden Chamber Invasion assay, another type of assays, as zymographic analysis or expression levels of biomarkers associated to cell invasion, should be performed, due to the limitations of the Boyden Chamber Invasion assay (Z.I. Stryker, Biomedicines 2019, doi:10.3390/biomedicines7020037; T.S. Gerashchenko, J. Clin. Med. 2019, 1092; doi:10.3390/jcm8081092).

Response: We thank the reviewer for sharing the two articles related to the use of the "Boyden Chamber for invasion assays." We agree with the reviewer that the MTT assay is not suitable to determine cell growth. We have changed "cell growth" to "cell viability" in figure 2B and in the text of the revised manuscript (Line 155 and 165), as well as discussed the limitation in the discussion section (lines 473-475).  We have also recognized the limitation of using the Boyden Chamber Invasion assay and EMT markers to conclude the role of MALAT1 in cell invasion. Given the current COVID-19 status in California (the labs and the University are shut down) and the timeline for the requested revision, we are not able to conduct more experiments with other methods. We have revised the statement in the results section (lines 155, 158). The limitation of the Boyden Chamber Invasion assay to determine cell invasion has been discussed in the Discussion section as well (lines 470-478). We also cited the articles provided by the reviewer as "new" references (references 54-55).

To confirm that MALAT1 is linked to trastuzumab resistance and to confirm that knockdown of MALAT1 increases sensitivity to trastuzumab, the authors should study the effects of MALAT1 knockdown on cell growth, cell invasion and EMT markers in sensitive and resistant trastuzumab cell models. Besides SKBR3/100-8 and BT474/100-2 cells, the authors should have applied MALAT1 knockdown on SKBR3 and BT474 cell models and compare the results to confirm that the MALAT1 knockdown effects were different or the same between sensitive and resistant cells and/or if the observed effects were specific of trastuzumab-resistance.

Response: We have conducted knockdown of MALAT1 in sensitive cell lines, SKBR3, and BT474. The effect of knockdown of MALAT1 on cell invasion was evaluated, and the data are presented in Figure 3C. After the knockdown of MALAT1, the number of invaded cells was reduced by 20% in SKBR3 cells and ≈11% in BT474 cells. The knockdown of MALAT1 in trastuzumab-resistant cells, SKBR3/100-8 and BT474/1002, did not reduce further the number of invaded cells (Figure 2C and 2D). However, once we knockdown MALAT1 in the resistant cells treated with trastuzumab, the cell invasion decreased significantly (Figure 2C and 2D). Hence the effect of MALAT1 knockdown in cell invasion was more effective in sensitive cells compared to resistant cells. Since the SKBR3 and BT474 cells are sensitive to trastuzumab treatment (Figure 2C and 2D), so we did not conduct additional trastuzumab treatment in the sensitive cells with knockdown of MALAT1. Our goal was to demonstrate that knockdown of MALAT1 in resistant model improves sensitivity to trastuzumab.

As we mentioned above, the resistant cell lines, SKBR3/100-8, and BT474/100-2 demonstrate an increase in EMT markers, and Wnt3 and β-catenin signaling.  The resistant cells also show increased cell invasion. Considering the role of MALAT1 in metastasis, we focused on the effect of knockdown of MALAT1 in the resistant model by examining cell invasion and changes in the EMT markers. We also tested the effects of MALAT1 knockdown in trastuzumab sensitive parental cells, SKBR3 and BT474 with and without Wnt3 treatment. Our previous study had shown that Wnt3 treatment reduced sensitivity to trastuzumab, with an increase in cell invasion and EMT markers (reference 44).

Nonetheless, we also evaluated the changes in EMT markers in response to MALAT1 knockdown in SKBR3 cells. The data was not provided in our earlier version of the manuscript. We have now added the data in the revised manuscript as supplemental figure S1. The Knock-down of MALAT1 in SKBR3 cells was correlated with EMT makers. In summary, MALAT1 knockdown had similar effects in both sensitive and resistant cells, concerning the downregulation of EMT makers. Additional changes are shown in the results section, lines 198-199.

We did not perform the MTT assay for SKBR3 cells with MALAT1 knockdown.  

The authors demonstrate in Figure 4 the protein expression levels of pAKT Ser473, T-AKT, and β-actin. Although, the authors did not say on figure legend the number of experiments performed. So densitometric analysis of at least three independent experiments should be performed and presented.

Response: The densitometric analysis for western blot data has been conducted, and the data has been presented in the bottom panels of Figures 4A and 4B. These analyses are from 4 independent experiments. More description has made in figure ledged of Figure 4A and 4B.  

Figure 4: Why the authors studied the relative expression of MALAT1 with or without MK-2206 and GDC0941 in JIMT1, MB231, and SKB3/AA cell models? Why the authors did not perform this assays in SKBR3/100-8, BT474/100-2, SKBR3, and BT474 cell models, if these cells correspond to sensitive and resistant trastuzumab cell models, in order to conclude that MALAT1 upregulation is mediated by PI3K/AKT pathway, being this mechanism linked to trastuzumab-resistance. Explanations of the selection of the cell lines should be performed.

Response: (1). The rationale to test the effects of MK-2206 (inhibitor of pan Akt, and sensitive to PIK3CA-mutant and cell lines with PTEN loss) and GDC0941 (PI3Kα/δ inhibitor) in MALAT1 expression in JIMT1, MB231, and SKBR3/AA cells was the following:

  • We wanted to test the effects of the inhibition of PI3K/Akt in MALAT1 expression in different HER2+ resistant cells, rather than just the two cell lines SKBR3/100-8 and BT474/100-2 cells. We believe the mechanism of resistance among JIMT1, SKBR/AA, SKBR3/100-8, and BT474/100-2 have some similarities but may also have differences. One common feature of all these cell lines is that they all express a higher level of pAkt, and all cell lines tested have upregulated MALAT1 (Figure 1 and Figure 4B). The differences are that with SKBR3/AA cells, these cells were transfected with active Akt, myr-Akt that resulted in high levels of active Akt protein (pAkt) expression, and subsequently showed decreased sensitivity to trastuzumab treatment. We have published the data in Cancer Research in 2010 (reference 43: Cancer Res. 2010, 70, 5475-5485). JIMT1 (ER-/HER2+) cells are generated from HER2+ breast cancer patients with known clinical resistance to trastuzumab. Although these cells also express a high level of pAkt, they may have other de novo mechanisms that contributed to trastuzumab resistance. SKBR3/100-8 (ER-/HER2+) and BT474/100-2 (ER+/HER2+) were generated by clonal selection from trastuzumab treatment and survival in our laboratory. These cell lines acquired trastuzumab resistance and showed activated Wnt3. 
  • We would also like to test if the regulatory role of PI3K/Akt pathway in MALAT1 expression is unique to HER2+ cells resistant to trastuzumab, or it has an effect on other breast cancer cells with active PI3K/Akt pathway. Hence the TNBC cells, MDA-MB231 with the most increase in MALAT1, were tested in this study.

 (2) We saw a similar effect of GDC0941 and MK-2206 from the data in figure 4C. Hence, we examined the effects of inhibition of pAkt by MK-2206 on MALAT1 expression in SKBR3/100-8 and BT474/1002 cell lines (Figure 4D).

(3) We did not overexpress MALAT1 again in the knockdown cells since we used siRNA knockdown, and those resistant lines were having a high level of MALAT1 already. We also did not use SKBR3 and BT474 cells to test the inhibition of pAkt on the effect of MALAT1 expression since the SKBR3 and BT474 cells without growth factor induction do not express a high level of pAkt (Figure 4A). Therefore, we feel using these two sensitive lines may not be relevant. However, in our further mechanisms' studies, we will use EGF to induce pAKt and test the effect of inhibiting PI3K/Akt in MALAT1 expression in SKBR3 and BT474 cells. Unfortunately, this will depend on when we get access back to our labs during this COVID-19 environment.   

In page 7, line 228, the authors say that "Inhibition of Akt by MK-2206 upregulated FOXO1 in SKBR3/100-8" but in BT474/100-2 cell model, what is the effect? Why the authors only used MK-2206 and did not apply GDC0941?

Response:  We only tested the inhibition of Akt by MK-2206 on FOXO1 expression in SKBR3/100-8 cells. This was to establish a direct link with FOXO1 in the regulation of the MALAT1 promotor via PI3K/Akt pathway shown in Figure 5B. We did not test the inhibition of Akt by MK-2206 on FOXO1 regulation in the current study. However, since our data shows that MK-2209 can inhibit pAKt, and downregulate MALAT1 expression in BT474/100-2 cells, we anticipate that the inhibition of Akt by MK-2206 will upregulate FOXO1 in BT474/100-2 cells as well.   

Why the authors selected the MB231 cell model for the in vivo studies and not used the other cell models generally applied in the previous in vitro assays? A justification should be provided.

Response: The MB231 cells showed the highest expression of MALAT1 and expressed EMT-like markers (high expression of Snail, Slug, Twist, and low expression of E-cadherin. These cells also have high expression of Nanog and CD44). MB231 cells also express high pAKt and low FOXO1. ChiP assay in Figure 4D demonstrated that overexpressing FOXO1 in MB231 increased the occupancy of FOXO1 in the promotor region of MALAT1 significantly. The data indicate that FOXO1 nuclear retention regulates MALAT1 expression. In addition, overexpressing FOXO1 in MD231 cells reduced the EMT-like transition. Since SKBR3, SKBR3/100-8, BT474, and BT474/100-2 did not form tumors in animal models from our previous studies. In contrast, MB231 cells proliferate in xenograft models, and these cells show the highest expression of MALAT1. Hence, we believe that MB231 cells are a suitable in vivo model to test the effect of MALAT1 on tumor formation.  

Based on the Author's data, in the discussion section, the authors should discuss the mechanisms by which targeting MALAT1 could overcome HER2+ breast cancer resistance to trastuzumab.

Response: We have demonstrated that the upregulation of MALAT1 results in poor or loss of response to treatment with trastuzumab for HER2+ cells. We have also shown that this process is associated with PI3K/Akt mediating FOXO1 nuclear retention, hence targeting the Akt pathway and stabilizing FOXO1 in the nucleus may be a strategy to control the overexpression of MALAT1. We have discussed this mechanism in the revised manuscript in the discussion section (lines 442-455, 462-469)  

Minor comments:

In figure 2B, y-legend, instead of Cell growth (%), the authors should write Cell viability (%), since MTT assay measures the reduction of a tetrazolium component (MTT) into an insoluble formazan product by the mitochondria of viable cells. Thus, MTT assay did not study cell growth but only cell metabolic activity or loss of viable cells.

Response: Thank you. The y-legend in figure 2B has been changed to "Cell Viability (%) in the revised manuscript.

Reviewer 3 Report

Comments for the authors:-

The current manuscript by Wu et al. comprises of characterizing the functional role of MALAT1, which is a long non-coding RNA (lncRNA) in aggressive breast cancers (BCs) such as TNBC and trastuzumab-resistant HER2 overexpressing BC cancers. They have demonstrated high expression of MALAT1 both at tissue and cell line level. They have performed both in vitro and in vivo experiments demonstrating the importance of MALAT1 in cell survival and cell invasion. At molecular level, they have demonstrated that loss of MALAT1 is associated with sensitization to trastuzumab therapy and leads to reversal of EMT gene signatures. They have also characterized the role of AKT is regulating MALAT1 in a FOXO1 dependent manner. Though the data demonstrated by authors are strong evidences of the function of MALAT1, the authors have represented a poorly constricted manuscript with very basic mistakes. The authors should be careful in preparation of manuscript as many of the figures are not correctly aligned. They lacked to demonstrate rescue experiments and response of MALAT1 with other HER2-targetted therapies. The experiments performed are repeated twice as indicated by standard deviation, which according to this reviewer should be represented as SEM with minimum three repeats with the respective technical repeats. Also, the figures presented are very hard to interpreted and lack consistency. The detail comments are as below:

  1. The introduction does not describe the link between AKT/FOXO1 and MALAT. This needs to be discussed in the introduction as it forms a major part of the second section of the manuscript.
  2. Why no IHC staining for MALAT1 was not performed on the BC tissues? How were the patients chosen and why no clinical data of these patients not discussed in the manuscript?
  3. The graphs represented in the manuscript are inconsistent. Why the labeling in the x-axis is not aligned with the bars of the sample represented?
  4. Figure 2: The graphs are not aligned with each other. Figure 2A is smaller than Figure 2B. The graphs of 2C and D have thicker axis than 2A and 2B?
  5. Figure 2A and B: Why TNBC lines were not used for SiRNA KD?
  6. Figure 2A and B: The authors must show rescue by exogenously overexpressing MALAT1 in these lines.
  7. Each graph should describe how the statistics was performed. It is interesting that most of the graphs demonstrate *p<0.05. The authors should carefully investigate their stats.
  8. The authors should challenge the HER2 lines with lapatinib and absence and presence of MALAT to highlight the importance of MALAT1 in determine the efficacy of HER2 dependent treatment. This experiment will also prove whether MALAT1 is specifically required for Trastuzumab sensitivity or overall HER2 dependent treatment sensitivity?
  9. Why stable MALAT1 KD cell were not developed and challenged with trastuzumab?
  10. The authors should also show colony formation assay using the resistant lines to demonstrate the anchorage-independent growth capability of these cells.
  11. Figure 3: Does exogenous overexpression of MALAT1 post siRNA transfection reinitiates EMT gene signature?
  12. Figure 3D: Why did the authors use MDA 231 cell lines here while these cells lines were not used in any of the previous experiments of Figure 2? The authors must logically explain the usage of the respective cell lines.
  13. Figure 3D: why the 231 cell were treated with SiRNA for 72 hours while other cells were treated for 24 hours?
  14. The authors lack to conclude the outcome of each result. This reviewer insists the authors to provide the outcome of every result they have demonstrated.
  15. P value: Generally p value is represented as **p<0.01 and *p<0.05. This reviewer strongly recommends that the authors should strictly follow this format of p value representation
  16. Figure 4D: why GDC0941 was challenged against the resistance lines?
  17. Line 228: the author use the word “unregulated”. They should use scientific terminology such as deregulated/downregulated.
  18. The author should describe the regulatory role of AKT on FOXO1 also at protein level. P-AKT goes inside the nucleus and phosphorylates FOXO1. This regulatory role must be described in details for readers. The authors should cite PMID: 29683733.
  19. Figure 4C: Again MDA 231 cells were used but not the HER2 cell lines. The authors should be consistent with their methodology.
  20. Figure 6A and 6B: The authors state that they have figure 6A and 6B. Labelling 6A and 6B are missing in the figure.
  21. The tumour growth: the data demonstrated by the authors of tumour growth shows no SEM/variation of tumour volume at respective time points especially in control. Interestingly, the tumour size shown in Figure 6C demonstrates very high level of difference between the tumour sizes. How is that possible?
  22. Figure 6C: The figure is blurred and author must increase its size for better clarity.
  23. Figure 6C: Why the authors have not shown representative images of mice from FOXO-18 cage.
  24. Figure 6D: the figure legend is confusing. Is the data representation of total RNA or MALAT1 expression?
  25. Figure 6D: If MALAT1 expression is represented, then this is data is obvious because the tumour size of the respective groups is significantly low.

Author Response

We wish to thank reviewer-3 for the helpful and constructive suggestions. Here are our responses to each reviewer-3 comments/suggestions.

Review 3

Comments and Suggestions for Authors

Comments for the authors:-

The current manuscript by Wu et al. comprises characterizing the functional role of MALAT1, which is a long non-coding RNA (lncRNA) in aggressive breast cancers (BCs) such as TNBC and trastuzumab-resistant HER2 overexpressing BC cancers. They have demonstrated high expression of MALAT1 both at tissue and cell line level. They have performed both in vitro and in vivo experiments demonstrating the importance of MALAT1 in cell survival and cell invasion. At the molecular level, they have demonstrated that loss of MALAT1 is associated with sensitization to trastuzumab therapy and leads to the reversal of EMT gene signatures. They have also characterized the role of AKT is regulating MALAT1 in a FOXO1 dependent manner. Though the data demonstrated by authors are strong evidence of the function of MALAT1, the authors have represented a poorly constricted manuscript with very basic mistakes. The authors should be careful in preparation of the manuscript, as many of the figures are not correctly aligned. They lacked to demonstrate rescue experiments and response of MALAT1 with other HER2-targetted therapies. The experiments performed are repeated twice as indicated by standard deviation, which, according to this reviewer, should be represented as SEM with a minimum of three repeats with the respective technical repeats. Also, the figures presented are very hard to interpret and lack consistency.

Response: We have followed the reviewer's comments and have revised the manuscript accordingly.

  • The figures are realigned.
  • The experiments were independently conducted four times in total. Each analysis was performed at least twice with four different independent times of sample collections, i.e., RNA was extracted from the cells from four different treatments. The cells used for each MTT and invasion assays were from four different cultures. The experiment conducted from the same samples’ preparation was also conducted at least two to three times. Three determinations for qPCR and invasion assays and 8 determinations for MTT were required at each performance. We have re-calculated the data as mean± SEM from the four independent experiments.
  • We have made significant revision for the manuscript and key parts of the revision are highlighted in yellow in the revised manuscript.

The detail comments are as below:

The introduction does not describe the link between AKT/FOXO1 and MALAT. This needs to be discussed in the introduction as it forms a major part of the second section of the manuscript.

Response: Thank you. We have added the description that shows the link between AKT/FOXO1 and MALAT1 in the introduction (lines 81-101) and discussion sections (lines 443-455) in the revised manuscript.

Why was no IHC staining for MALAT1 not performed on the BC tissues? How were the patients were chosen and why no clinical data of these patients not discussed in the manuscript?

Response: Since MALAT1 is a non-coding RNA, we did not use IHC staining for MALAT1 in breast cancer tissues. Instead, we conducted the FISH analysis with the MALAT1 probe to compare its expression between breast cancer tissues (n=18) and tissues from benign breast tumors (n=3) at the beginning of the study. The data from FISH analysis also showed upregulation of MALAT1 in breast cancer tissues, and the high expression of MALAT1 was seen in TNBC and HER2+ breast cancer patients. To identify genes associated with the risk of breast cancer and drug resistance, we have started sequencing the tissues from breast cancer patients and patients with benign disease in our Lab. The MALAT1 data in breast cancer tissues were obtained for the sequencing data in the Core.

Breast cancer tissues and non-cancer tissues used in this study were obtained from the breast tissue bank at the Integrated Clinical and Tissue Biobank/Biorepository Core in our cancer center. The tissues in the breast tissue bank were the majority from African American and Hispanic/Latina women living in South Los Angeles in California. These patients were either examined in our Breast/Mammography Clinic or treated in the Hematology/Oncology Clinic at the Martin Luther King Ambulatory Care Center associated with our University. The clinical information was collected for each patient. All breast cancer patients have been followed up for at least for 5-years or more. We have added a detailed description in the Methods section (lines 518-524).

Since the sample size is small, we included all available data for the study. The purpose was to compare MALAT1 expression between cancer and non-cancer tissues/patients. Further research with a large sample size is required. Since the breast study is ongoing, and we continue sequencing analysis, we will conduct a retrospective study with sufficient study power to validate the prognosis and prediction value of MALAT1 in breast cancer patients. 

The graphs represented in the manuscript are inconsistent. Why is the labeling in the x-axis not aligned with the bars of the sample represented?

Response: We have changed the x-axis labeling in Figure 2C and 2D to keep it consistent with other graphs.

Figure 2: The graphs are not aligned with each other. Figure 2A is smaller than Figure 2B. The graphs of 2C and D have a thicker axis than 2A and 2B?

Response: We have changed the x-axis labeling in Figure 2C and 2D to keep it consistent with other graphs. The graphs are now justified and aligned consistently.

Figures 2A and B: Why TNBC lines were not used for siRNA KD?

Response: The data in Figure 2 is to shows that knockdown of MALAT1 increases or restores the sensitivity of trastuzumab treatment in HER2+ breast cancer cells. The TNBC line with the siRNA knockdown of MALAT1 data is shown in Figure 3D to confirm that knockdown of MALAT1 reduces EMT makers in both HER2+ and TNBC cells.  

Figures 2A and B: The authors must show rescue by exogenously overexpressing MALAT1 in these lines.

Response: We agree with the reviewer that if we conducted overexpression of MALAT1 in SKBR3 and examined its response to trastuzumab, that would be more convincing. Due to the timeline for revision and the current COVID-19 situation in Los Angeles, California, we are not able to conduct this experiment in our Lab. Access to our research labs and the University is restricted. Only urgent and essential activities are allowed. We will perform the transfection experiments in our future studies according to the reviewer's suggestion once the Lab is reopened.  

Each graph should describe how the statistics was performed. It is interesting that most of the graphs demonstrate *p<0.05. The authors should carefully investigate their stats.

Response: The methods of statistics have been added in the figure legends in each figure in the revised manuscript. The significances were demonstrated as “*p<0.05” if the P-value is less than 0.05, and “**p<0.01” if the P-value is less than 0.01 in figures.   

The authors should challenge the HER2 lines with lapatinib and absence and presence of MALAT to highlight the importance of MALAT1 in determining the efficacy of HER2 dependent treatment. This experiment will also prove whether MALAT1 is specifically required for Trastuzumab sensitivity or overall HER2 dependent treatment sensitivity?

Response: We thank the reviewer for this excellent suggestion. Yes, as soon as we can get back to our research labs, we will challenge the HER2 cells with lapatinib in the presence and absence of MALAT1. These studies will also be done with cell lines with either MALAT1 overexpression and/or knockdown.

Why stable MALAT1 KD cell were not developed and challenged with trastuzumab?

Response: We did not perform stable MALAT1 KD since we considered all experiments are conducted in 2-3 days. The stable MALAT1 KD will take a long time for the selection process with a positive knockdown.   

The authors should also show colony formation assay using the resistant lines to demonstrate the anchorage-independent growth capability of these cells.

Response: We have recognized the limitation by using only the MTT assay to assess the cell growth in this study since MTT is more likely to demonstrate cell viability. In the revised manuscript, we have changed the y-Axis in Figure 2B labeling as "Cell Viability (%)" instead of "Cell growth (%)", as well as in the text of revised manuscript (highlighted in yellow) The limitation has been discussed in the discussion section of the revised manuscript (lines: 473-478).   

Figure 3: Does exogenous overexpression of MALAT1 post siRNA transfection reinitiates EMT gene signature?

Response: We did not perform exogenous addition or overexpression of MALAT1 in the cells we investigated. This is a very interesting suggestion, and we will perform these studies when we get back to our labs.

Figure 3D: Why did the authors use MDA 231 cell lines here while these cell lines were not used in any of the previous experiments of Figure 2? The authors must logically explain the usage of the respective cell lines.

Response: As we mentioned above, the study is focused on understanding the role of MALAT1 in HER2+ cells that are clinically resistant to trastuzumab, or made resistant from a clonal selection of parental lines exposed to trastuzumab. Figure 2 shows the knockdown of MALAT1 in trastuzumab-resistant cells, decreased cell viability, and inhibited cell invasion while restoring the sensitivity to trastuzumab. Figure 3 shows that MALAT1 knockdown downregulates EMT markers in trastuzumab-resistant HER2+ cells; otherwise, those cells would have increased EMT and cell invasion. The TNBC line with the siRNA knockdown of MALAT1 in Figure 3D confirmed that knockdown of MALAT1 could decrease EMT markers in both HER2+ and TNBC cells. Suggesting, that the role of MALAT1 in inducing EMT may not be unique to HER2+ cells, but could affect other breast cancer cells. We have made the clarification in the text of revised manuscript (Lines: 208-209, yellow highlighted).  

 Figure 3D: why the 231 cells were treated with SiRNA for 72 hours while other cells were treated for 24 hours?

Response: All cell cultures were treated with siRNA for 72 hours. Subsequently, when HER2+ cells were co-treated with trastuzumab or Wnt3, we first treated these cells with siRNA for 24 hours. We then added trastuzumab or Wnt3 in the medium containing siRNA for additional 48 hours. The overall siRNA treatment was maintained for 72 hours.    

The authors lack to conclude the outcome of each result. This reviewer insists the authors provide the outcome of every result they have demonstrated.

Response: We have concluded the outcome of each result in the main text, as well as in all figure legends in the revised manuscript. The changes are heighted in yellow.

P value: Generally p value is represented as **p<0.01 and *p<0.05. This reviewer strongly recommends that the authors should strictly follow this format of p value representation

Response: Thank you. We have revised the expression of the p-value. It is now represented as **p<0.01 and *p<0.05 in the revised manuscript. 

Figure 4D: why GDC0941 was challenged against the resistance lines?

Response: We saw a similar efficacy of GDC0941 and MK-2206 from the data in figure 4C. Therefore we examined the effect of inhibition of pAkt by MK-2206 on MALAT1 expression in SKBR3/100-8 and BT474/1002 cell lines only (Figure 4D).

Line 228: the Authors use the word "unregulated." They should use scientific terminology such as deregulated/downregulated.

Response: We have corrected the spelling, "unregulated" has been corrected as "upregulated."

The Author should describe the regulatory role of AKT on FOXO1 also at protein level. P-AKT goes inside the nucleus and phosphorylates FOXO1. This regulatory role must be described in details for readers. The authors should cite PMID: 29683733.

Response: Thank you for referring to this essential publication. We have described the regulation of AKT on FOXO1 at the protein level and pAKt mediating nuclear export of FOXO1 in the introduction (lines: 89-101) and the discussion section (lines: 443-455) in the revised manuscript. The article PMID: 2938733 has been cited as reference 37. 

Figure 4C: Again, MDA 231 cells were used but not the HER2 cell lines. The authors should be consistent with their methodology.

Response: We first tested the efficacy of the two inhibitors in inhibiting MALAT1 to establish the link between MALAT1 and PI3K/Akt pathway in HER2+, HER2+ with myr-Akt transfection and TNBC cell lines (all lines have been tested to overexpressing pAKt). The data in figure 4C showed that all three cells response to both inhibitors and decreased MALAT1. Therefore, we conducted the inhibition experiments in the two trastuzumab-resistant HER2+ cell lines by using the pen Akt inhibitor, MK-226, only in figure 4D.

Figures 6A and 6B: The authors state that they have figures 6A and 6B. Labelling 6A and 6B are missing in the figure.

Response: Thank you for noting this error. Figure 6A and 6B labeling have been added.

The tumour growth: the data demonstrated by the authors of tumour growth shows no SEM/variation of tumour volume at respective time points especially in control. Interestingly, the tumour size shown in Figure 6C demonstrates very high level of difference between the tumour sizes. How is that possible?

Response:  The two smaller tumors in the control group were detected after four weeks, and one was found after five weeks. The study was ended at week six since most of the tumors in the control group had reached the IACUC's protocol for tumor size guidelines. We did not include these three smaller tumors for calculation of tumor volume for the control group in figure 6B in the original manuscript.

The data has been carefully replotted (all 18 tumors have been included in the calculation) and presented as figure 6C (left-penal in the revised manuscript) to address the reviewer's concerns. The average tumor volume was slightly reduced, as shown in the revised figure 6C (left). The calculations for FOXO1-18 and FOXO1-4 were included in all tumors in the previous version of the manuscript. We have made the error bars more visible in the graph in the revised manuscript.

Figure 6C: The figure is blurred, and the Author must increase its size for better clarity.

Response: We have increased all fonts of labeling in the figures and reformatted all figures.  The figures have been saved in higher resolution in zip files and provided separately to the Journal.  

Figure 6C: Why the authors have not shown representative images of mice from FOXO-18 cage.

Response: The images for the FOXO-18 group were not taken for each mouse. The sizes of tumors from FOXO-18 were slightly bigger than the tumors from FOXO-4 at the end of the study. But the tumor formation in both FOXO1-18 and FOXO1-4 groups were fewer. We document every tumor for the FOXO1-4 group. Hence the images from the FOXO1-4 group are presented (In the revised manuscript the data is in Figure 6B).

Figure 6D: the figure legend is confusing. Is the data representation of total RNA or MALAT1 expression?

Response: We isolated and purified total RNA from tumors and then conducted RT-qPCR with MALAT1 primer. Hence the data presented show mRNA expression of MALAT1. We have edited the most figure legends in the revised manuscript.

Figure 6D: If MALAT1 expression is represented, then this is data is obvious because the tumour size of the respective groups is significantly low.

Response: The data indicates that high MALAT1 promotes tumor growth, and downregulation of AKT inhibits tumor growth in vivo (RNA samples used for RT-qPCR in the different groups were added at equal amounts)

Round 2

Reviewer 2 Report

The authors performed the requested revisions to the manuscript. I consider that the manuscript is scientifically accurate. However, the English should be improved, since there are several grammatical errors, reason why I strongly recommend revising the manuscript thoroughly prior to any publication. Minor revisions for English grammar are mandatory.

Author Response

Response:

We have performed English checking and corrected grammars in the manuscript. 

Reviewer 3 Report

This reviewer is happy with the author's reply to all the comments.

One critical suggestion, the Figure 6C. The graph needs to be represented as SEM but not SD.

Author Response

The Figure 6C. The graph needs to be represented as SEM but not SD.

Response: We have revised the figure 6C, the graph has been represented as SEM.